# First Principle Surface Analysis of YF_3_ and Isostructural HoF_3_

**DOI:** 10.3390/ma15176048

**Published:** 2022-09-01

**Authors:** Jennifer Anders, Niklas Limberg, Beate Paulus

**Affiliations:** Institute for Chemistry and Biochemistry, Freie Universität Berlin, Arnimallee 22, 14195 Berlin, Germany; niklas.limberg@fu-berlin.de (N.L.); b.paulus@fu-berlin.de (B.P.)

**Keywords:** geochemical twins, REE, HFSE, waimirite, DFT, DFT+U, Hubbard, surface energy, Wulff plots

## Abstract

The trifluorides of the two high field strength elements yttrium and holmium are studied by periodic density functional theory. As a lanthanide, holmium also belongs to the group of rare earth elements (REE). Due to their equivalent geochemical behavior, both elements form a geochemical twin pair and consequently, yttrium is generally associated with the REE as REE+Y. Interestingly, it has been found that DFT/DFT+U describe bulk HoF_3_ best, when the 4f-electrons are excluded from the valence region. An extensive surface stability analysis of YF_3_ (PBE) and HoF_3_ (PBE+U
${}_{d}$
/3 eV/4f-in-core) using two-dimensional surface models (slabs) is performed. All seven low-lying Miller indices surfaces are considered with all possible stoichiometric or substoichiometric terminations with a maximal fluorine-deficit of two. This leads to a scope of 24 terminations per compound. The resulting Wulff plots consists of seven surfaces with 5–26% abundance for YF_3_ and six surfaces with 6–34% for HoF_3_. The stoichiometric (010) surface is dominating in both compounds. However, subtle differences have been found between these two geochemical twins.

## 1. Introduction

Yttrium and holmium form a geochemical twin pair. The term emphasizes their identical geochemical behavior caused by the equal ratio of charge to radius in their only stable oxidation state +III. According to their small ionic radii of 1.075 Å (Y) and 1.072 Å (Ho) in nine-fold coordination [1] and their high oxidation state, both belong to the interesting group of high field strength elements (HFSE). As a lanthanide, holmium also belongs to the rare earth elements (REE). Due to their twin character, yttrium is also often associated with that group [2,3,4].

As fluorides, both metals can be used for different specific applications. The wide-band-gap material YF_3_ has very good properties for laser applications [5,6,7,8]. Doped with trivalent REE cations, YF_3_ is also applicable as an optical filter in 157-nm photolithography [9]. Another emerging field of application is solid-state fluoride batteries, resulting from the very high conductivity of fluoride anions [10,11,12,13,14]. HoF_3_ is interesting for magnetic high-field applications as, e.g., a contrast agent, due to the very high magnetic moment of holmium [15,16]. Moreover, YF_3_ and HoF_3_ are important precursors for the synthesis of the respective pure metallic compounds [17,18]. In nature, YF_3_ is found within the mineral waimirite-(Y), which contains high concentrations of other REE [8]. Fluoride plays a significant role in accumulating HFSE and REE within hydrothermal fluids, as these cations do not form such stable complexes with chloride [2,19,20,21]. Interestingly, those fluoride-rich hydrothermal fluids produce ores with a non-chondritic excess of yttrium over holmium. It is suggested that one underlying reason for the twin separation is their different affinity to fluorine, which was found in dissolving experiments of YF_3_ and HoF_3_ in diluted hydrofluoric acid [20]. To lay one foundation for future quantum chemical studies on the different fluorine-affinity of yttrium and holmium, we started with an investigation of the respective trifluorides and their surfaces.

In accordance with their twin behavior, solid YF_3_ and HoF_3_ occur in the same crystal structure type of 
β
-YF_3_ (space group *Pnma*, fully occupied Wyckoff positions (Y) 4c, (F) 4c, (F) 8d (Figure 1)) [18,22,23]. This is their only stable phase up to 1343 K (HoF_3_) or 1350 K (YF_3_), which is well beyond the temperature regime of hydrothermal fluids of typically 323–873 K [7,18,24,25]. The ionic radii of the middle and late lanthanides Sm(III)–Lu(III) differ by only ≤6 pm compared to Y(III) [1]. Consequently, all their trifluorides crystallize as well in 
β
-YF_3_ [18,22,23,26,27,28,29,30,31]. The same low-temperature phase is also found for the two actinides Bk(III) and Cf(III) [32,33,34]. Due to the often observed analogy of actinides to lanthanides, the same crystal structure is assumed for the, so far, experimentally unknown heavier actinide trifluorides of Es(III)–Lr(III) [35]. Interestingly, the known orthorhombic low-temperature phase of plutonium trihydride is also reasoned to be an exotic example of a 
β
-YF_3_ structure [36]. Outside the f-block, the 
β
-YF_3_-structure is experimentally known for bismuth trifluoride [23,37,38] and predicted as an accessible meta-stable phase for the trichlorides of Y(III) and Bi(III) [38].

To the best knowledge of the authors, no first principle surface stability analysis of any compound within the whole structure type is available in the literature. The only surface calculation of any compound of 
β
-YF_3_ structure was published in 2013 by Ye et al. [39] on two selected surfaces of DyF_3_ (001) and (101). They only calculated the surfaces matching their experimentally obtained nano-plates. However, these and other experiments on this class of compounds clearly demonstrate that the obtained surface structures are very dependent on the experimental conditions, especially on the utilized nature and geometry of the substrate, as well as on the solvent and fluoride concentration [14,16,40,41,42,43,44,45]. The present work analyzes the inherent quantum chemical stability of all of the seven low Miller indices (
hkl
) surfaces, namely (001), (010), (100), (011), (101), (110) and (111). A previous study on another metal trifluoride, AlF
${}_{3}$
 revealed stoichiometric or substoichiometric surfaces with a small fluorine-deficit as the most stable terminations [46]. Additionally, a substoichiometric fluorine content has also been found for YbF_3_ thin films made from ion assisted deposition [47]. Consequently, this study includes all possible stoichiometric terminations and those with a small-to-moderate fluorine-deficit of 1–2 fluorine atoms per surface unit cell. This results in a scope of 24 terminations. The obtained surface energy results are combined with the geometry of the surface cut by a Wulff analysis to examine the expected surface abundance [48,49].

## 2. Methodology

### 2.1. Computational Details

All calculations were performed in the Vienna Ab Initio Simulation Package (VASP, version 5.4.4) [50] on the supercomputer cluster HLRN in Berlin and Göttingen, Germany. using periodic density functional theory (DFT) with a generalised gradient approximation (GGA). As an exchange–correlation functional, the one of Perdew–Burke–Ernzerhof (PBE) is applied [51]. The inner shell electrons were described by the projector augmented wave (PAW) method [52,53]. The outer shell electrons were expanded in plane waves.

For converged YF_3_ total bulk energies, the VASP potential files F_h (“hard”, 7 electrons) and Y_sv (11 electrons) were applied together with a 
9×9×9
 Monkhorst–Pack grid. In accordance with the F_h potential file, a kinetic energy cut-off of 772.6 eV was used. For HoF_3_, both available Ho potential files Ho_3 (9 electrons, 4f-in-core) and Ho (21 electrons, 4f-in-valence) were evaluated with respective grid sizes of 
7×7×7
 and 
3×3×3
. On holmium, the Hubbard-type correction in the simple Dudarev formalism was applied [54]. In a test series of 1–10 eV in 1 eV steps with U
${}_{d}$
 (with Ho_3) and U
${}_{f}$
 (with Ho), PBE+U
${}_{d}$
 with 3 eV agreed best with the crystal structure and the presumed electronic structure (Table 1 and Appendix A). As an electronic structure reference, bulk HoF_3_ was also calculated with the Heyd–Scuseria–Ernzerhof hybrid functional (HSE06) [55].

For electron smearing, tests on several bulk and slabs structures of both trifluorides were performed, comparing Gaussian smearing with the tetrahedron method with Blöchl correction [56]. No energy difference within the applied self-consistent field (SCF) convergence criteria could be found. We therefore used Gaussian smearing on our insulating trifluorides.

Apart from the trifluorides, molecular fluorine, as well as metallic yttrium and holmium, were also considered. The first was calculated in a cubic box of 25 Å length. For the latter two, Gaussian smearing could not be applied. A convergence test with 1st and 2nd-order Methfessel–Paxton smearing with widths of 0.05–0.35 eV yielded 2nd-order Methfessel–Paxton smearing with a width of 0.10 eV (Y) or 0.15 eV (Ho) as the best combination to minimize the difference between total energy and free energy.

Each bulk structure started from the respective, experimental crystal structure (YF_3_ [23], HoF_3_ [22], Y [57], Ho [58]) and was fully relaxed in atomic positions, lattice constants and volume. The accurate precision setting was applied. As convergence criteria, 0.01 meV per unit cell was used for SCF total energies and 0.1 meV per unit cell for the difference in total energy between two ionic steps. Final total energies, density of states (DOS) and Bader charges were performed with an SCF criteria of 0.001 meV. All DOS plots and Bader charges, as well as all HoF_3_ data, were calculated with allowed spin polarization. To aid SCF convergence, an additional support grid (.ADDGRID.) and/or a reduced minimal mixing parameter for Kerker’s initial approximation [59] (AMIN) of <0.01 were applied on most slabs.

Symmetric slabs were built from the relaxed bulk structure with the Python package pymatgen [60,61]. The vacuum height perpendicular to the surface was tested for one stoichiometric termination of YF_3_ (001). The converged value of 25 Å was applied for all slabs. For slab calculations, only one *k*-point was used perpendicular to the surface. For the other two directions, we applied the same *k*-point grid size as in bulk. The complete slabs were relaxed in atomic positions.

DOS plots and band structures were generated with pymatgen. Wulff plots were constructed with the WulffPack Python package [62]. Atomic structures were visualized in VESTA [63].

### 2.2. Choice of Electronic Structure Method

The effect of dispersion was tested by applying Grimme’s dispersion correction with Becke–Johnson damping (D3(BJ)) [64]. From PBE to PBE+D3(BJ), the lattice constants changed only by 1.9–4.5 pm or 0.3–1.0% during the full optimization of atomic positions, lattice constants and volume of YF_3_. Due to this small deviation, we neglected dispersion correction for our highly ionic systems.

For HoF_3_, a test series was performed to decide whether to treat the 4f-electrons inside the core or at the valence level. Hubbard-type Coulomb parameters of 1–10 eV were scanned for the 4f-in-core with U
${}_{d}$
 acting on Ho-d orbitals, as well as for 4f-in-valence with U
${}_{f}$
 acting on Ho-f orbitals. It should be noted that the Ho-5d orbitals mainly constituted the broad conduction band in both approaches. Yet, they also hybridized in the valence band mainly constructed by F-2p (Appendix A). The PBE+U benchmark plots for unit cell parameters and band gaps are given in the SI with further discussion (Appendix A). All HoF_3_ (PBE+U
${}_{d}$
/4f-in-core) band structures resembled the YF_3_ (PBE) one and produced comparable F-2p to Ho-5d or Y-4d charge transfer band gaps of 7–8 eV (Appendix A). By adding exact exchange via HSE06/4f-in-core, these bands were further separated to 11 eV. Whereas, HSE06/4f-in-valence predicted an Ho-4f to Ho-4f transition of 8 eV. In contrast, PBE/4f-in-valence was not able to separate the partially filled 4f
${}^{10}$
 into un-/occupied bands. Instead, it placed the Fermi-level (
$E_{F}$
) inside the 4f band, predicting a pseudo-metal. When introducing the additional Coulomb potential of 1–10 eV onto the 4f in PBE+U
${}_{f}$
, this 4f–4f gap was tuneable from 1 eV to a maximum of 6 eV. At U
${}_{f}\geq 5$
 eV, the nature of the band gap changed to a charge transfer of F-2p to Ho-4f. Unfortunately, no measured band gap exists in the literature for HoF_3_. Therefore, it was not possible to pin-point the true band gap, nor to evaluate the correct nature of that transition. Nevertheless, based on a purely empirical model derived from other lanthanide compounds, HoF_3_ is expected to have a band gap of ca. 9 eV [65]. This empirically estimated band gap, as well as the calculated HSE06 reference, were best reproduced without including the 4f-electrons explicitly.

Another quantity upon which to judge the applied electronic structure method was the Bader charges obtained by applying the atoms in molecules (AIM) population analysis [66,67,68,69,70]. For both bulk materials of YF_3_ and HoF_3_, all tested methods predicted a metal charge of 2.2–2.4 e and fluorine charge of −(0.7–0.8) e. The Bader charges of all applied methods with 4f-in-core or valence agreed well with each other and thus suggested that including 4f explicitly was not necessary for HoF_3_.

Furthermore, all methods used with 4f-in-valence predicted a high-spin bulk unit cell with all four holmium aligned resulting in an electronic magnetic moment of 16 µ
${}_{B}$
. This ferromagnetic result was obtained even when starting from anti-ferromagnetic spin arrangements. According to the experimentally known magnetic structures, the physically correct spin arrangement is anti-ferromagnetic below 
0.53
 K or paramagnetic above [71].

To summarize, not including the 4f-electrons explicitly provided the best electronic structure results. The differences between simple PBE and PBE+U
${}_{d}$
 were minor. When considering the unit cell parameters given in Table 1, PBE+U
${}_{d}$
/3 eV/4f-in-core performed best with deviations of as little as 0.1–0.8%.

### 2.3. Surface Energy

The surface formation energy (
$E_{\mathrm{surf}}$
) is generally calculated from the total energy of the 2D-periodic slab (
$E_{n}$
), the energy of the 3D-periodic bulk unit cell (
$E_{\mathrm{bulk}}$
) and the surface area of the slab (*A*):
(1)
$$E_{\mathrm{surf}}^{\mathrm{bd}}=\frac{E_{n}-nE_{\mathrm{bulk}}}{2A}.$$


*n* is the slab thickness measured in unit cells. We label this bulk-derived surface energy 
$E_{\mathrm{surf}}^{\mathrm{bd}}$
. Equation (Equation 1) is used for all YF_3_ surface energies. In this work, we also considered surfaces with a substoichiometric amount of fluorine. For these, the fluorine potential µ
${}_{F}$
 for each missing fluorine was added to the numerator of Equation (Equation 1). µ
${}_{F}$
 was obtained from 
$E_{\mathrm{bulk}}$
, the bulk energy per atom of the pure metal of yttrium or holmium (µ
${}_{M}$
), as well as the number of metal (
$n_{M}=4$
) and fluorine (
$n_{F}=12$
) atoms within the bulk MF_3_ unit cell:
(2)
$$\mu_{F}=\frac{E_{\mathrm{bulk}}-n_{M}\mu_{M}}{n_{F}}.$$


Yet, as pointed out by Boettger, this bulk-derived surface energy (
$E_{\mathrm{surf}}^{\mathrm{bd}}$
) can lead to diverging 
$E_{\mathrm{surf}}$
 with respect to *n* [72]. This can be avoided by using slab-derived (sd) energies only:
(3)
$$E_{\mathrm{surf}}^{\mathrm{sd}}=\frac{E_{n}-n\left(E_{n}-E_{n-1}\right)}{2A}.$$



$E_{\mathrm{bulk}}$
 is then replaced by the difference of 
$E_{n}$
 to the total energy of the next smaller slab (
$E_{n-1}$
). For HoF_3_, we indeed observed linearly diverging 
$E_{\mathrm{surf}}^{\mathrm{bd}}$
 when applying Equation (Equation 1), despite system sizes of up to 7 UC or Ho_28_F_84_. Depending on the (
hkl
), this stoichiometry corresponds to 12, 24 or 26 HoF_3_-layers. Likely, this is a result of the allowed spin-polarization with Hubbard-type correction and atomic relaxation of the whole slab. It can be seen in Appendix A, that this linear divergence only appears after relaxation in 
$E_{\mathrm{surf},\mathrm{opt}}^{\mathrm{bd}}$
. The unrelaxed surface energies 
$E_{\mathrm{surf},\mathrm{SP}}^{\mathrm{bd}}$
 show no divergence. In YF_3_, no divergent 
$E_{\mathrm{surf},\mathrm{opt}}^{\mathrm{bd}}$
 are observed. Here, no Hubbard-type correction is applied and the atomic relaxation is performed without spin polarization. A comparison of slab convergence by both equations is given in Appendix A. Due to the divergence issue, all HoF_3_ surface energies given within this paper are slab-derived using Equation (Equation 3), which nicely converge. As each 
$E_{\mathrm{surf}}^{\mathrm{sd}}$
 is derived from two slabs differing by one unit cell in size, at least three slabs are needed to determine convergence. Whereas for 
$E_{\mathrm{surf}}^{\mathrm{bd}}$
, these are just two. Due to the observed convergence of 
$E_{\mathrm{surf}}^{\mathrm{bd}}$
 in YF_3_, only two slab thicknesses are modeled for many terminations. Therefore, the convergence of the respective 
$E_{\mathrm{surf}}^{\mathrm{sd}}$
 cannot be evaluated. As a consequence, we used the converged 
$E_{\mathrm{surf}}^{\mathrm{bd}}$
 for YF_3_ to compare with the converged HoF_3_ 
$E_{\mathrm{surf}}^{\mathrm{sd}}$
. All YF_3_ bulk-derived surface energies converged within 0.03 J m
${}^{-2}$
 at slab thickness of about 5–5.5 UC or 10–22 YF_3_-layers (Appendix A). The HoF_3_ slab-derived surface energies of 14 terminations, including all of the most stable ones per Miller indices, converged to 0.01 J m
${}^{-2}$
 or less within a slab thickness of about 6–6.5 UC or 12–26 HoF_3_-layers (Appendix A). Some of the higher energy terminations converged only to 0.02–0.04 J m
${}^{-2}$
 at that thickness, while four high energy terminations did not converge even to 0.1 J m
${}^{-2}$
. Fortunately, it is clear from their surface energies that even within the present uncertainty, those high energy terminations do not compete with the lowest energy ones. The slab thickness convergence for HoF_3_ is visualized by error bars in Appendix A.

## 3. Results and Discussion

### 3.1. Bulk Properties

For YF_3_, the PBE relaxed lattice constants, given in Table 1, agree very well with both experimental values, which are underestimated by as little as 0.5–1.5% [22,23]. The resulting unit cell volume is underestimated by 2.6–2.7%, which is still in good agreement for a GGA functional. The best performing HoF_3_ method against the only available experimental unit cell data and the calculated HSE06 band gaps was found to be PBE+U
${}_{d}$
/3 eV/4f-in-core. The resulting unit cell parameters deviate by as little as 0.1–0.8%.

The respective F–M bond length on the PBE (YF_3_) and PBE+U
${}_{d}$
 level (HoF_3_) are 
$R_{F-Y}=\left\{2.28;2.29;2.46\right\}$
 Å and 
$R_{F-\mathrm{Ho}}=\left\{2.30;2.32;2.45\right\}$
 Å. These agree perfectly with the measured interatomic distances of 2.3–2.6 Å [22,23].

Before we come to the surfaces, we evaluate possible energetic differences between the two geochemical twins as bulk materials. We calculate the electronic contribution to the formation enthalpies (
$\Delta H_{f}$
) according to:



$$2M{}_{\left(s\right)}+3F{}_{2}{}_{\left(g\right)}\overset{2\;\Delta H_{f}}{\to }2\mathrm{MF}{}_{3}{}_{\left(s\right)}$$



The electronic energies are taken from the bulk metals in hcp (*P*63/*mmc*) structure and the bulk trifluorides, as well as molecular fluorine. For YF_3_, we obtained an electronic contribution of 
−1591.1
 kJ mol
${}^{-1}$
 versus 
−1587.3
 kJ mol
${}^{-1}$
 for HoF_3_. Thus, judged by the electronic energies only, both trifluorides are equally strong bound with a very small favor of 
−3.7
 kJ mol
${}^{-1}$
 or 0.2% for YF_3_ over HoF_3_.

### 3.2. Surface Energies

The surface energies of all calculated terminations are given in Table 2. The given metal surface coordination number (CN
${}_{\mathrm{surf}}$
) is determined with a bond length cut-off of 
$R_{F-M}\leq$
 2.60 Å. Table 2 also includes the nominal net surface charge (
$q_{\mathrm{surf}}$
) caused by substoichiometric fluoride. Finally, the last column includes the surface abundance for each respective most stable termination predicted by Wulff construction (%
${}_{\mathrm{surf}}$
).

The two terminations, (110)-1 and -2 greatly illustrate the importance of atomic relaxation of the surface prior analysis. Before relaxation, nothing but the very surface layer differs within each (
hkl
) cut. As both terminations are stoichiometric, they are also identical in composition. However, for both trifluorides, the unrelaxed (110)-1 surface is by 0.6 J m
${}^{-2}$
 more stable than the one of (110)-2 (see 
$E_{\mathrm{surf}}^{\mathrm{unrel}.}$
 in Table 2). When allowed to relax in atomic positions, the {5,9,8} surface coordinations of (110)-2 rearrange into {6,8,8} Appendix A. Hence, the surface energy reduces by as much as 1.41 J m
${}^{-2}$
 for YF_3_ or 1.18 J m
${}^{-2}$
 for HoF_3_. In contrast, termination (110)-1 already starts at a higher surface coordination of {6,9,8}, before it also rearranges into {6,8,8}. According to the lesser degree of rearrangement, its surface energy only reduces by 0.79 J m
${}^{-2}$
 for YF_3_ or 0.60 J m
${}^{-2}$
 for HoF_3_. Both rearranged terminations are structurally equivalent.

The argumentation in CN
${}_{\mathrm{surf}}$
 cannot only be applied to explain the high 
$E_{\mathrm{surf}}^{\mathrm{unrel}.}$
 of some terminations, but is also partially applicable to the relaxed 
$E_{\mathrm{surf}}$
. Within all YF_3_ (
hkl
) subsets, except those of (101) and (111), the respective smallest CN
${}_{\mathrm{surf}}$
 value correlates with 
$E_{\mathrm{surf}}$
. Thus, the smaller the smallest coordination polyhedron, the less stable the surface and the higher its surface energy. For example, a 4-fold coordination, as present in many surfaces with a fluorine-deficit of two, is only found for the highest 
$E_{\mathrm{surf}}$
 within the (
hkl
) subset. Yet, this correlation holds only for the very minimal value within a set of CN
${}_{\mathrm{surf}}$
. No correlation can be found for the remaining, higher CN
${}_{\mathrm{surf}}$
 values of the same surface. Therefore, these cannot explain the energetic order of two terminations showing the same smallest CN
${}_{\mathrm{surf}}$
 value (as seen e.g., in YF_3_ (100)-1 and -3 in Table 2). For HoF_3_, this correlation of surface coordination and stability has two more exceptions. Here, the stability of the least and second-least stable (100) and (001) terminations flip compared to YF_3_, without any change in CN
${}_{\mathrm{surf}}$
. Prior to surface relaxation, all coordination polyhedrons of YF_3_ and HoF_3_ are identical as they share the same crystal bulk structure. After relaxation, this is still true for twenty terminations (Appendix A). Only four rearranged terminations differ slightly in surface coordination between YF_3_ and HoF_3_. All of these four terminations belong to the less stable surfaces within the respective (
hkl
). All most or second-most stable terminations are identical in surface coordination between YF_3_ and HoF_3_. The most stable surface structure termination for each of the seven Miller indices is shown in Figure 2.

As shown in Figure 3, the obtained 
$E_{\mathrm{surf}}$
 are similar in magnitude and, within most Miller indices, the order of terminations is equal between YF_3_ and HoF_3_. Within convergence, this is also true for the two stoichiometric terminations of (110) and (101), which are very similar in surface energy. For (100) and (001), the least and second-least stable terminations switch their order between YF_3_ and HoF_3_. Here, HoF_3_ prefers the surface with a nominal surface net charge of 
+2
 over the stoichiometric one. The only difference in termination order between the two compounds, which also affects the most stable surface, is found in (111). For YF_3_, the most stable surface is (111)-2, which shows a surface coordination of CN
${}_{\mathrm{surf}}=\left\{6,5,8,8\right\}$
. Whereas, HoF_3_ prefers (111)-3 with CN
${}_{\mathrm{surf}}=\left\{6,7,6,9\right\}$
 (Figure 2). However, both of these terminations are equal in constitution with a fluorine-deficit of 1 per surface.

Even though the order within one (
hkl
) is largely the same between YF_3_ and HoF_3_, the order between the different (
hkl
) does change. For YF_3_, the overall two most stable surfaces are (010)-1 and (001)-2, which are both stoichiometric and give a surface energy of 0.58 J m
${}^{-2}$
. This is closely followed by the stoichiometric surface (011)-2. Medium stable surfaces are found for (101)-3 and (111)-2, which are both substoichiometric surfaces missing a single fluorine. The two least stable surfaces (110)-1/-2 and (100)-2 prefer a stoichiometric termination again.

(010)-1 is also the overall most stable surface for HoF_3_, but the remaining surfaces differ in order. (100)-2, which is the most unstable (
hkl
) in YF_3_, is the second-most stable one in HoF_3_. The moderately stable surfaces (001)-2, (011)-2 and (101)-3 have equivalent surface energies within the slab thickness convergence of 0.01 J m
${}^{-2}$
. (111)-3 is the second least stable surface. The least stable surface is stoichiometric (110)-1/-2, which is not even part of the Wulff plot (Figure 4).

From the corresponding energies of the respective most stable surfaces shown in Figure 2, the Wulff plots are constructed. A Wulff plot visualizes the thermodynamically most stable crystal shape at quantum chemical conditions of 0 K and vacuum. To test the dependence of surface ratios on the slab thickness convergence, an estimation on the maximum possible error is given in the SI (Appendix A).

The largest surface area of over one quarter in YF_3_ or one third in HoF_3_ is formed by (010). That HoF_3_ prefers (010) even stronger is a consequence of its surface energy being 0.11 J m
${}^{-2}$
 more stable than any other. This contrasts YF_3_, for which (001) has the same surface energy as (010), as well as a very closely (0.03 J m
${}^{-2}$
) following (011). Nonetheless, the geometric interdependence of surfaces cause a much smaller abundance of only 10% for (001) versus more than double for (011) in YF_3_. The third most abundant surface in YF_3_ is (101) with 20%, which is one of the two obtained substoichiometric surfaces. As these same three surfaces (001), (011) and (101) are only medium stable in HoF_3_, they also only constitute 6%, 13% and 14% of the overall surface. The second substoichiometric surface present in both Wulff plots is (111), which forms an area of 10% in YF_3_ and 7% in HoF_3_. Thus, almost one third of the YF_3_ crystal is made from terminations with a nominal positive net charge of +1. Whereas for HoF_3_, these are only about a fifth. The two least stable surfaces of YF_3_ are (100) and (110), which constitute about 7% and 5%. In HoF_3_, the latter is to such an extend energetically unstable, that it is completely excluded from the Wulff plot. (100), on the other hand, turns to be the second most stable and second most abundant surface in HoF_3_. It constructs one quarter of the total surface.

The overall scale of most stable surface energies per Miller indices is comparable between YF_3_ and HoF_3_. The respective ranges are 0.58–1.03 J m
${}^{-2}$
 for YF_3_ and 0.47–1.00 J m
${}^{-2}$
 for HoF_3_. However, the resulting average by the Wulff plot is 16% higher for YF_3_ with 
$ØE_{\mathrm{surf}}=0.70$
 J m
${}^{-2}$
 than for HoF_3_ with 
$ØE_{\mathrm{surf}}=0.59$
 J m
${}^{-2}$
. This means, that forming surfaces from the bulk crystal involves a higher thermodynamic barrier in YF_3_ than in HoF_3_. This is interesting, as there is no significant difference within the formation enthalpies of the bulk. The difference in average surface energy also hints, that the thermodynamic barrier of crystal nucleation is also higher in YF_3_ than in HoF_3_. Though, to accurately predict the nucleation, the nature of the respective precursors and the media needs to be considered.

#### 3.2.1. Bader Charges

Figure 5 shows the partial charges obtained by Bader analysis for all 24 thickness-converged slabs of YF_3_ or HoF_3_. As these slabs are built from up to 104 atoms, a large number of very similar charges are obtained. To ease comparison, the Bader charges given in Figure 5 are rounded to 0.1 e. The Bader charges of the central slab atoms reproduce the bulk values with 2.4 e for yttrium, 2.3 e for holmium, as well as 
−0.8
 e for fluorine in both compounds. Only HoF_3_ (011)-3 shows a marginally increased central slab value of 2.4 e for holmium. In general, the Miller indices do not seem to affect the Bader charges within the analyzed accuracy. The highly ionic partial charge on fluorine does not significantly change for any slab with an overall range of −(0.7–1.0) e. Moreover, all stoichiometric surfaces have practically the same metal partial charges with 2.3–2.4 e. On the contrary, substoichiometric surfaces with a fluorine-deficit of one do all have at least one metal center charged less at the surface. These might be as low as 1.7 e, as found for the third termination of (100) in both trifluorides. Subsequently, surfaces with a fluorine-deficit of two contain even less ionic metal centers at the surface. The least charged metal center is observed again in (100) with only 1.2–1.3 e by the forth termination. The Bader charges suggest that all substoichiometric surfaces missing two fluorine have at at least a single surface metal center in an oxidation state of +II.

#### 3.2.2. Surface Band Gaps

Investigating the electronic properties, we found that the band gap, total DOS and projected DOS of atoms central within the slab converged already at the smallest slab thickness. All band gaps are plotted in Appendix A. For the most stable termination of each (
hkl
), the DOS near the Fermi level are also shown in Appendix A. The surface DOS narrow the band gap within all terminations. For stoichiometric slabs, the direct band gap is reduced from the bulk value of ca. 8 eV to 4–7 eV. Thus, all stoichiometric surfaces remain fully within the insulating regime. In contrast, for substoichiometric terminations, the direct band gap collapses to 0–1 eV, predicting a pseudo-metallic or narrow-band-gap surface. For HoF_3_ (101) and (111), only one spin direction shows a nearly metallic character. The other stays insulating (5–6 eV). It should be noted that this pseudo-metallic or narrow-band-gap electronic structure at the substoichiometric surfaces might be strongly effected by the chosen neutral 2D-periodic model. However, as this paper is focusing on the relative stability of surfaces, we are not investigating the nature of the band gaps, nor the observed surface magnetism or spin-asymmetric band gaps of some HoF_3_ surfaces further.

## 4. Conclusions

The aim of this study was to obtain the relative surface stabilities, in order to find the most abundant surfaces of the two REE trifluorides, YF_3_ and HoF_3_ according to their inherent quantum chemical stability. While YF_3_ can be treated on the DFT level, the 4f-electrons of HoF_3_ required an extensive electronic structure benchmark evaluating DFT, DFT+U and hybrid DFT against the crystal unit cell, band gaps and Bader charges. On the DFT or DFT+U level, our results show that including the 4f-electrons explicitly within the plane wave expansion worsens the geometrical and band gap results, while the Bader charges stay unaffected. Considering also the experimentally not observed high-spin preference of the 4f-electrons, as well as the strongly increased computational demand, we treated HoF_3_ by a 4f-in-core DFT+U approach, in which the Hubbard-type correction is applied on the Ho-5d orbitals, which mix into the valence band mainly constructed by F-2p.

From the relaxed bulk, surface models were created for any of the seven low-lying Miller indices. Our analysis included all possible stoichiometric terminations, as well as those showing a small to moderate fluorine-deficit. The surfaces were quantified by Bader charges, band gaps and DOS. From the resulting scope of 24 surfaces, we constructed the first Wulff plots for the whole class of 
β
-YF_3_-structured compounds.

We found that, within each Miller indices, both trifluorides prefer the same termination with the exception of (111), in which different surface coordinations are favored.

Comparing the different Miller indices, both compounds clearly show stoichiometric (010) as the most stable surface. The preference of the other surfaces, though, varies between the two. The greatest difference is found for (100), which is the second-most stable surface for HoF_3_, but the second-least stable one for YF_3_. On average, the surface energy predicted by the Wulff plot is higher for YF_3_ than for HoF_3_. This suggests a higher thermodynamical barrier for the formation of YF_3_ surfaces from the bulk.

In total, one third of the predicted equilibrium crystal shape of YF_3_ is made from the substoichiometric terminations of (101) and (111) missing a single fluorine per surface. In HoF_3_, these only constitute a fifth. In the search for the underlying reason between the different fluorine affinity of the two compounds, this different availability of substoichiometric surfaces is an interesting finding. However, to evaluate possible effects, further studies are needed that actually model binding interactions with these surfaces. These should also apply more elaborate binding analysis tools than simple population analysis.

## Figures and Tables

**Figure 1 materials-15-06048-f001:**
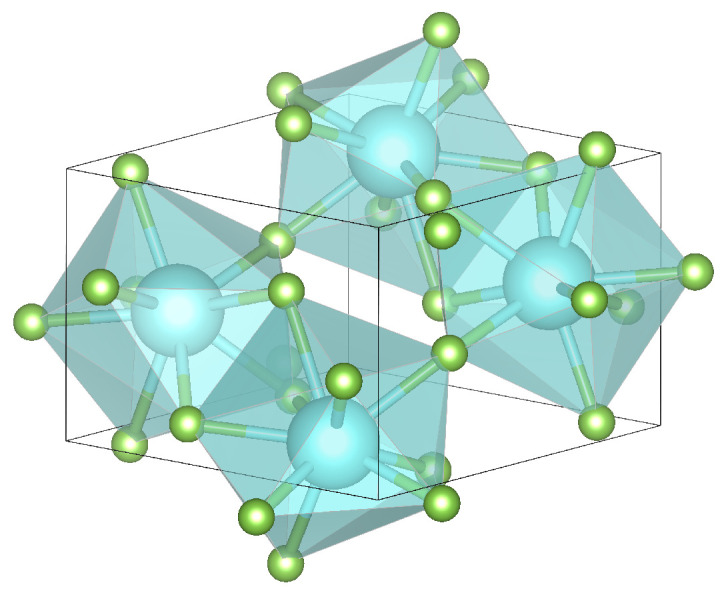
The orthorhombic unit cell of 
β
-YF_3_ with *Pnma* symmetry. Lattice constants are given in Table 1. The distorted tricapped trigonal prisms formed by nine fluorides (green) around each yttrium (cyan) are visualized by transparent, cyan planes.

**Figure 2 materials-15-06048-f002:**
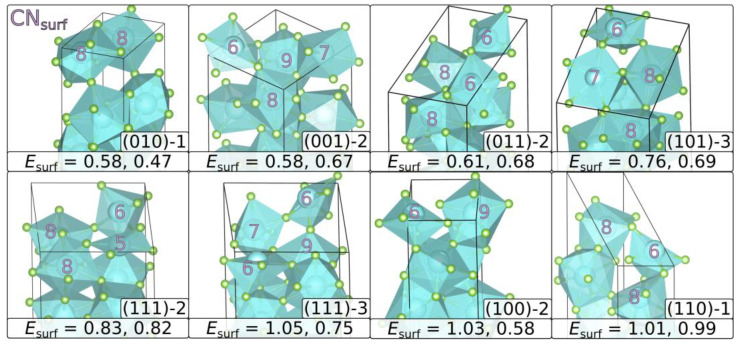
Most stable terminations of the relaxed surface structures: the coordination number of the surface metals (CN_surf_) and the surface energies in J m^−2^ (*E*_surf_) are given. The first entry corresponds to YF_3_ and the second to HoF_3_. The mean of both values corresponds to the given order from top left to bottom right. Each (*hkl*) slab is rotated in a way to show the surface coordination best. For (111), two surfaces are given, as (111)-2 is preferred by YF_3_ and (111)-3 by HoF_3_.

**Figure 3 materials-15-06048-f003:**
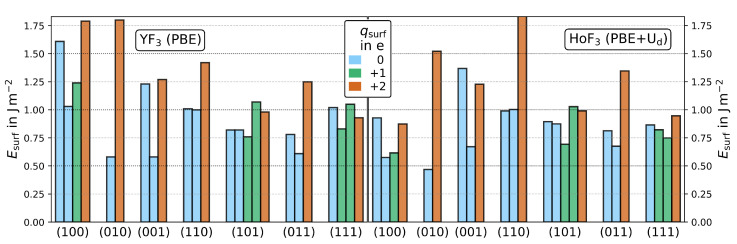
Relaxed surface energies of YF_3_ (left, PBE) and HoF_3_ (right, PBE+U
${}_{d}$
/3 eV/4f-in-core) of all 24 terminations. The surfaces are color-coded to their nominal surface net charge (
$q_{\mathrm{surf}}$
) in e of 0 (blue), 
+1
 (green) or 
+2
 (orange). This magnified plot does not show the HoF_3_ (110)-3 value of 2.09 J m
${}^{-2}$
.

**Figure 4 materials-15-06048-f004:**
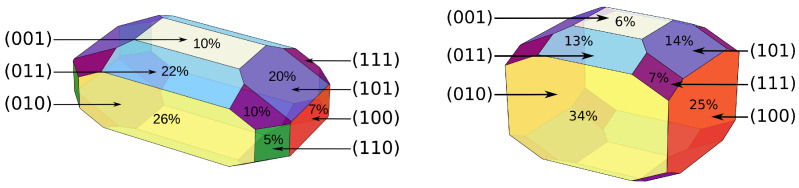
Wulff Plots of YF_3_ (left, PBE) and HoF_3_ (right, PBE+U
${}_{d}$
/3 eV/4f-in-core) from relaxed surfaces. The percentage shows the relative abundance of each surface, which is also given in Table 2.

**Figure 5 materials-15-06048-f005:**
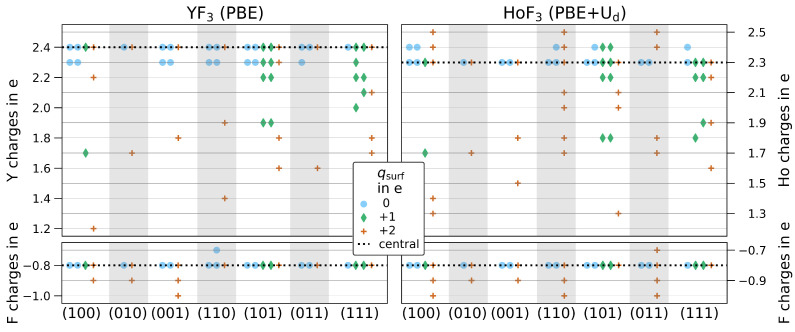
Bader charges rounded to 0.1 e for all slabs of YF_3_ (left, PBE) and HoF_3_ (right, PBE+U
${}_{d}$
/3 eV/4f-in-core): the Bader charges of the bulk are highlighted by the dotted line (Y: 2.4 e, Ho: 2.3 e, F: 
−0.8
 e). Terminations are differently colored by their formal surface net charge (
$q_{\mathrm{surf}}$
) in e for 0 (blue cycle), 
+1
 (green diamond) and 
+2
 (orange cross).

**Table 1 materials-15-06048-t001:** Comparison of the relaxed unit cells to experiment including standard deviation in parentheses [22,23]. Given is the absolute difference (
Δ
), as well as the deviation from experiment in percentage (
$\Delta_{\%}$
).

	YF_3_ (PBE)				HoF_3_ (PBE+U ${}_{d}$ /3 eV/4f-in-Core)			
	a **(** **Å** **)**	b **(Å)**	c **(Å)**	V **(Å** ${}^{3}$ **)**	a **(Å)**	b **(Å)**	c **(Å)**	V **(Å** ${}^{3}$ **)**
calc.	6.3215	6.8059	4.3300	186.29	6.4164	6.8796	4.3440	191.76
exp. [22]	6.353(3)	6.850(3)	4.393(3)	191.2	6.404(3)	6.875(3)	4.379(3)	192.8
Δ	0.032	0.044	0.063	4.9	0.012	0.005	0.035	1.0
$\Delta_{\%}$	0.5%	0.6%	1.4%	2.6%	0.2%	0.1%	0.8%	0.5%
exp. [23]	6.3537(7)	6.8545(7)	4.3953(5)	191.42				
Δ	0.0322	0.0486	0.0653	5.13	—			
$\Delta_{\%}$	0.5%	0.7%	1.5%	2.7%				

**Table 2 materials-15-06048-t002:** The YF_3_ (PBE) and HoF_3_ (PBE+U
${}_{d}$
/3 eV/4f-in-core) surfaces with respective terminations (term.), slab thickness in layers of formula units without terminal F-deficit (
$L_{\mathrm{MF}3}$
), nominal surface net charge (
$q_{\mathrm{surf}}$
) in e, surface energies of relaxed (
$E_{\mathrm{surf}}$
) and unrelaxed slabs (
$E_{\mathrm{surf}}^{\mathrm{unrel}.}$
) in J m
${}^{-2}$
, as well as the relaxed surface metal coordination number (CN
${}_{\mathrm{surf}}$
). The lowest surface energies per (
hkl
) cut are highlighted in bold. For these, also the abundance obtained by the Wulff plot (%
${}_{\mathrm{surf}}$
) is given.

			$L_{\mathrm{MF}3}$		CN ${}_{\mathrm{surf}}$		$E_{\mathrm{surf}}$ ( $E_{\mathrm{surf}}^{\mathrm{unrel}.}$ )		% ${}_{\mathrm{surf}}$	
**(** hkl **)**	**term.**	$q_{\mathrm{surf}}$	**YF_3_**	**HoF_3_**	**YF_3_**	**HoF_3_**	**YF_3_**	**HoF_3_**	**YF_3_**	**HoF_3_**
(100)	1	0	20	24	5,9		1.61 (2.87)	0.93 (1.48)		
	2	0	22	26	6,9		**1.03** (2.02)	**0.58** (0.96)	7%	25%
	3	+1	20	24	5,8		1.24 (1.61)	0.62 (0.68)		
	4	+2	22	26	4,7		1.79 (2.14)	0.87 (0.90)		
(010)	1	0	10	12	8,8		**0.58** (0.84)	**0.47** (0.49)	26%	34%
	2	+2	10	12	6,6		1.80 (2.05)	1.52 (1.52)		
(001)	1	0	20	24	5,8,8,9		1.23 (2.45)	1.37 (2.25)		
	2	0	22	26	6,7,8,9		**0.58** (1.39)	**0.67** (1.16)	10%	6%
	3	+2	22	26	4,5,8,9		1.27 (1.70)	1.23 (1.29)		
(110)	1	0	20	24	6,8,8		**1.01** (1.80)	**0.99** (1.59)	5%	0%
	2	0	22	26	6,8,8		**1.00** (2.41)	**1.00** (2.18)		
	3	+2	22	26	4,6,9	4,6,8	1.42 (1.73)	2.09 (1.36)		
(101)	1	0	20	24	6,7,8,8		0.82 (1.48)	0.89 (1.33)		
	2	0	20	24	6,6,8,8		0.82 (3.34)	0.88 (3.17)		
	3	+1	20	24	6,7,8,8		**0.76** (1.16)	**0.69** (0.89)	20%	14%
	4	+1	22	26	5,6,7,9	5,6,8,8	1.07 (2.10)	1.03 (1.70)		
	5	+2	20	24	4,5,8,8	5,6,8,8	0.98 (1.39)	0.99 (0.99)		
(011)	1	0	10	12	7,7,9,9		0.78 (1.30)	0.81 (1.14)		
	2	0	10	12	6,6,8,8		**0.61** (1.32)	**0.68** (1.15)	22%	13%
	3	+2	10	12	4,4,8,8		1.25 (1.68)	1.35 (1.38)		
(111)	1	0	20	24	6,7,7,8	7,7,8,8	1.02 (3.46)	0.87 (3.29)		
	2	+1	20	24	5,6,8,8		**0.83** (1.30)	0.82 (1.04)	10%	
	3	+1	22	26	6,6,7,9		1.05 (1.70)	**0.75** (1.11)		7%
	4	+2	20	24	5,5,7,7		0.93 (1.22)	0.95 (1.13)		

## Data Availability

See Appendix A. Further data can be requested from the authors.

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
