# Peer review of "First Principle Surface Analysis of YF3 and Isostructural HoF3"

_materials, 2022, doi:10.3390/ma15176048_

Round 1
Reviewer 1 Report
This paper systematically investigated the electronic structure of YF3 and HoF3. Both Pnma (room-temperature phase) bulk crystals and surfaces are studied. Various parameters including choice of k-point mesh, pseudopotentials, Hubbard-U value, exchange-correlation functional, and dispersion correction have been evaluated. All significant parameters have been considered and the choice is completely appropriate for these two crystals.
It is out of expectation that including f-electrons in Ho would yield worse results for lattice constant and band gap. There are some minor issues mentioned below. I suggest the paper be published ASAP once the author finished the corrections. The paper is written in high quality and could guide all future studies about YF3 and HoF3 surfaces.
1. Could the author mention the thickness of the slabs?
2. Is there any atom fixed during the relaxation of slabs? It is better to mention it in the methodology section.
3. The F_h potential is usually used for high pressure or F2 dimer calculations. What is the reason for using F_h potential here? Does give a better result than normal F potential?
4. Since BJ damping is included, I suggest directly addressing it as D3BJ without the bracket. So the functional can be written as PBE-D3BJ.
5. The spin configuration is not clear here. Did the author try the anti-ferromagnetic initial configuration (negative and positive MAGMOM)? Was it finally optimized to a ferromagnetic configuration? (Line 151)
Author Response
We thank the reviewer for taking the time to evaluate our manuscript. We are delighted about your positive feedback and very grateful for your detailed suggestions/questions. We have included your suggested improvements nr. 1, 2 and 5 in the revised version of our manuscript. All changes are marked in red. 1. Thank you for pointing out, that the slab size is not so clearly stated in the main paper. We have added the number of formula unit layers to the respective lines 183,198,201 in the Methodology, as well as in the overview Table 2. 2. The middle part of the slabs were not fixed. We added this in line 118. 3. Thank you for the detailed question. We have chosen the hard potential file of fluorine, as we planned from the beginning to perform adsorption studies of aqueous HF onto the obtained surfaces. As the bond distance of H-F is very short and to keep a consistent set-up during the whole project, we used F_h already from the beginning. In initial tests, we found a bond elongation of 0.6 pm for HF in a vacuum box, when using the usual potential files (F,H: 93.77 pm) instead of F_h & H_h (F_h,H_h: 93.15 pm). As high level calculations and experimental values on NIST agree on ca. 91.7 pm, the _h value performs a little better. Comparing the potential files on an actual adsorption, we tested a YF3-(100) supercell of (2x3x2)UC with a single HF adsorbed. On the YF3 surface, the difference is very subtle, but due to the difference in total energy of free HF, the non-hard potential files give a 3 kJ/mol higher adsorption energy. 4. Despite your sound argument for removing the parenthesis in "D3(BJ)", for style consistency with other papers within our group, we would like keep it. We would also like to point out, that even papers published with Prof. Dr. Grimme as corresponding author use the parenthesis notation. [https://doi.org/10.1002/cphc.201100521|https://doi.org/10.1002/jcc.21759] 5. Thank you once more for pointing this ambiguity out. We changed line 159 accordingly. Yes, we have tried anti-ferromagnetic spin-arrangements of the four Ho-centers within the bulk unit cell in any permutation and with different absolute values. We did not test bigger supercells. However, experiments on HoF3 by Brown et al. (https://doi.org/10.1088/0953-8984/2/19/013) have shown, that the magnetic unit cell is not bigger than the chemical one. They suggest an arrangement of [++--] with 5.7 Bohr magneton (nuclear+electronic) on each Ho(III) along 66degree to [001]. We did also not try non-collinear calculations. Apparently, on the DFT level, non-collinear, spin-orbit coupled calculations are needed to describe the magnetic moment of the 4f-electrons in HoF3 correctly. That being said, applying this set-up on the much larger surface models, would have been out of our computational reach.Reviewer 2 Report
This work "First Principle Surface Analysis of β-YF3-Structured Rare Earth Element Trifluorides" is submitted to Materials journal MDPI. Jennifer Anders and other authors report novel results on ab initio optimization and VASP-PAW modeling of the YF3 and HoF3 surfaces. There are interesting and profound results for PBE/PBE+Ud/4f-in-core calculations, Bader charges, Wulff plots, etc. The stoichiometric (010) surface was found as dominating in both compositions. The U ~ 1–10 eV change is discussed, 4f-in-valence/4f-in-core are compared, bulk and surface properties are thoroughly compared.
I have several comments on this manuscript:
1) The calculations were done for YF3 and HoY3 bulk and surfaces. The title should be modified appropriately to include both these compounds. It is too general in the current form, just 1 (Ho) of 14 rare earth elements is actually investigated.
2) In lines 149-157: the authors explain their choice of the PBE+Ud/3 eV/4f-in-core method for the surface, this can be mentioned in abstract too.
3) Figure 2 should be modified to show full plots.
4) In line 338: "project" should be "study"
5) Supplementary Materials are not included to review. In fact, all listed Figures S1-S6 and Tables S1-S6 are worth to be reported.
The manuscript is worth publishing in Section "Materials Chemistry" of Materials.
Author Response
We thank the reviewer for taking the time to evaluate our manuscript. We are very grateful positive feedback and for your suggestions. We have included your suggested improvements nr. 1, 2 and 4 in the revised version of our manuscript. All changes are marked in red.
1: According to your first comment, we adapted the title to: "First Principle Surface Analysis of YF3 and Isostructural HoF3"
2: Thank you for your suggestion. We added the more detailed methods to the abstract.
3: In the current form, Figure 2 shows the most stable surface termination of the respective (hkl) slab. We thoroughly thought about how to visualize the different surfaces best. As the corresponding, symmetric slabs a very large (5-5.5UC = 10-22 YF3-layers = Y20F60-Y22F66 for YF3 and 6-6.5UC = 12-26 HoF3-layers = Ho24F72-Ho26F78 for HoF3), we cannot show the whole slabs without loosing focus on the 2-4 metal centers at the surface. However, to make the slab sizes more clear, we have added the number of formula unit layers to the respective lines 183,198,201 in the Methodology, as well as in the overview Table 2.
4: This has been updated in line 355.
5: We are not fully sure, we understood your 5th comment correctly. The SI should have been available for the review process and it will be published alongside the main paper to be accessible to the reader. As the main paper has already a considerable length of 14 papers, we would like keep the 10 pages of SI outside the main paper. Especially the long slab thickness convergence tables would worsen the readability.